# Shades of Gratitude: Exploring Varieties of Transcendent Beliefs and Experience

Pamela Ebsytne King *, Rebecca Ann Baer, Sean A. Noe, Stephanie Trudeau, Susan A. Mangan and Shannon Rose Constable

Thrive Center for Human Development, Fuller Theological Seminary, 180 N. Oakland Avenue, Pasadena, CA 91182, USA
* Correspondence: pamking@fuller.edu

**Abstract:** The study of gratitude has expanded beyond interpersonal gratitude and considers how people respond to gifts that are not caused by human agency. Given the discord between the prominent understanding of gratitude requiring the appropriate recognition of a gift to a giver and the increasing divergence of transcendent belief systems that do not acknowledge a transcendent or cosmic giver, we explored how people with different worldviews viewed and experienced gratitude. Transcendence does not hinge on metaphysical beliefs, but it can be experienced phenomenologically and subjectively. We conducted a case-study narrative analysis ($N = 6$) that represents participants from three different categories of belief systems: theistic, non-theistic but spiritual, and other. Our findings demonstrate how people link their transcendent narrative identity to their thoughts, feelings, and behaviors pertaining to gratitude. Although the theistic participants thanked God for gifts, others who experienced transcendence without a clear referent or source described responding to gratitude by sharing goodness forward. These narratives suggest that the recognition and appreciation of a gift stemming from beyond human cause may be enough to generate transcendent emotions and values that prompt beyond-the-self behaviors.

**Keywords:** gratitude; transcendent gratitude; cosmic gratitude; transcendence; beliefs; virtue; spirituality; religion; case study; qualitative

## 1. Introduction

Given the mounting evidence in support of gratitude's positive effects on well-being and social ties, it is not surprising that the study of gratitude continues to expand, moving beyond examining gratitude between and toward people to examining gratitude toward more transcendent sources. This expansion is evident in recent publications on gratitude to less tangible benefactors (Tsang et al. 2021), transpersonal gratitude (Hlava et al. 2014; Steindl-Rast 2004), gratitude to God (Krause et al. 2014, 2015, 2017; Nelson et al. 2022; Roberts 2014; Rosmarin et al. 2011; Wilt and Exline 2022), higher-order gratitude (Lin 2014, 2017), existential gratitude (Jans-Beken and Wong 2021), and cosmic gratitude (Cohoe 2022; Manela 2019; Roberts 2014). Within this study, we were interested in how people's transcendent beliefs are linked to their views and experiences of gratitude for gifts that are not attributed to human agency. Although people in the US are moving away from conventional religious self-identification and affiliation toward broader understandings of spirituality and transcendence (Ammerman 2013; Kim et al. in press; Smith 2021; Steensland et al. 2018), psychological research generally neglects nuances of transcendent construals. Consequently, we conducted an in-depth case-study analysis of six individuals with either theistic, non-theistic but spiritual, or other beliefs.

### 1.1. The Power of Transcendent Narratives

Based on McAdams and Pals (2006) theory of personality, all persons internalize narratives that are more or less coherently integrated within their identities. These nar-

rative identities serve "to provide the person's life with some degree of unity, purpose, and meaning" (McAdams and Pals 2006, pp. 209–10). More recent work extends this conceptualization of narrative to distinguish transcendent narrative identities in order to highlight the unique orienting power of types of narratives imbued with sacred significance informing virtues such as gratitude and motivating moral actions (King 2020; King et al. 2020; Schnitker et al. 2019). Research points to the importance of religious traditions and institutions that offer transcendent narratives to promote meaning making and purpose (see Damon 2008; Furrow et al. 2004; King et al.; Lapsley and Hardy 2017; Liang and Ketcham 2017). However, not all persons have identities informed by clear conceptualizations of God or the divine. As noted above, trends show that fewer Americans ascribe to traditional theistic views about the transcendent (Smith 2021). Existing research demonstrates that theistic and nontheistic transcendent experiences may shape the meaning-making processes that galvanize a sense of self and worldview (Kim et al. in press; King et al. 2014; Mahoney et al. 2021; Nelson et al. 2022). People are increasingly adopting narratives about other sources of ultimacy or of the sacred that do not always include a metaphysical reality. Many are wrestling with what they believe, frequently departing from traditional faith, and their conceptualizations are often nebulous and unformed.

Increasingly, people report experiences of transcendence that do not involve a supernatural entity (Kim et al. in press; Nelson et al. 2021; Yaden et al. 2021). Callaway et al. (2020) describe that people not only conceptualize transcendence within theistic, humanistic, and naturalistic worldviews, but may experience ontological, phenomenological, or subjective transcendence. All of these offer a sense of expansion beyond the self as a way to find meaning. Ontological transcendence involves a metaphysical or supernatural reality. Within phenomenological transcendence, expansion beyond the self occurs in finding deep connection and obligation to an entity that is necessary for the conditions of life, such as humanity or nature. Still, for others, the self becomes the conduit for a sense of expansion that provides meaning or, in some cases, results in nihilism. Subjective transcendence may occur through the resonance and emotions inspired through art and beauty or through one's own agency. In the cases of phenomenological and subjective transcendence, individuals might have clear belief systems, but those systems do not explain a distinct agentic cosmic giver. Regardless of the type of transcendence, the beliefs or the experience of transcending the self defines ultimacy, is luminous, and defines what is essential or sacred-like.

Research suggests that clear views of the sacred and ultimate have orienting power on people's lives, yet many people are unclear about what beliefs and experiences define their lives. Emmons' (1999) research on strivings demonstrates that ultimate concerns serve to organize entire goal systems and orient life aims. In addition, life purposes are more likely to be incorporated into one's narrative identity when they include transcendent, spiritual, or sacred content (Bronk 2014; Damon 2008). Mundane goals given sacred meaning are pursued with greater effort, provide more meaning, and receive more social support than unsanctified goals (Mahoney et al. 2021). Thus, a transcendent narrative identity has organizing power and serves to instantiate meaning and motivates the development of virtues such as gratitude, a sense of belonging, and purpose beyond one's own life that inspire fidelity to one's beliefs, and a life lived accordingly (King 2020; Schnitker et al. 2019). Even narratives that are not embedded within religious contexts can serve to "sanctify" the beliefs, attitudes, and actions that comprise virtues. Given that increasing research demonstrates that people's narratives and sources of meaning do not necessarily stem from metaphysical beliefs but can be shaped by varied experiences of transcendence, within this study, we pursued *transcendent gratitude*. We were interested in exploring the self-reported transcendent narratives of participants and how these come to bear on experiences and practices of gratitude.

### 1.2. Gratitude

Since the turn of the century, research on interpersonal gratitude has continuously demonstrated gratitude's powerful effects on individual and interpersonal flourishing

(Emmons 2019). Within the existing psychological research, gratitude has been conceptualized as an affective state, meaning that gratitude is felt temporarily. as well as a trait, or an aspect of one's personality that endures over time (Carlisle and Tsang 2013; Emmons and McCullough 2003; McCullough et al. 2001, 2002; Lin 2014), as a life orientation (Wood et al. 2010), and as a virtue (DeSteno et al. 2019; Merçon-Vargas et al. 2017; Tudge et al. 2018). Despite the dissent on how to classify gratitude, generally, within the research literature, gratitude is most often understood as a response to a specific benefit received from a tangible *human* benefactor. Evolutionary theories within psychology emphasize the importance of the dynamic of exchange within interactions serving to bind communities together (McCullough et al. 2008). Such theories provide support for this binding function of gratitude by theorizing that gratitude produces a prosocial dynamic by alerting individuals to the prosocial efforts of others, while promoting them to behave prosocially in turn.

Recently, however, research has begun to highlight that experiences of gratitude towards more intangible agents might differ from interpersonal gratitude (Tsang et al. 2021). Such conceivable intangible benefactors could include entities such as governments; corporations; or more religious/spiritual agents such as God, spirits, ancestors, etc. Bolstering the case for the need to examine the linkages between more intangible benefactors and gratitude responses, early research on Gratitude to God demonstrated positive effects on well-being *beyond* the effects of interpersonal gratitude (Krause et al. 2014, 2015; Knabb et al. 2021).

In this study, we set out to explore this newer area of gratitude research by examining how transcendent narratives and belief systems shape experiences of gratitude toward intangible benefactors that might be considered divine, sources of transcendence, or other representations of ultimate reality. We chose to conceptualize gratitude as a virtue. Virtues involve constellations of thoughts, feelings, and behaviors that enable a person to reliably do what is right or good and are embedded in transcendent narratives (King 2020). Consequently, we define gratitude as the virtue consisting of habituated patterns of thinking, feeling, and behaving that allow an individual to reliably receive and respond to gifts and inspire a gracious response to the giver and/or to others.

*1.3. Transcendent Gratitude*

Transcendent-narrative identities are beyond-the-self belief systems that are informed by transcendent experiences that provide orienting power that directs purpose and meaning making, and this, in turn, motivates the development of virtues. Within this framework, we were interested in exploring *transcendent gratitude*, which we conceptualized as a virtue rooted in the transcendent narrative identities of individuals and prompted by a response to gifts that cannot plausibly be attributed to human agency. While interpersonal gratitude is generally based on a benefactor giving a gift that is received, recognized, and can be responded to, not all conceptions of transcendence, such as the connectedness of all beings, are personified, nor do they have motivations or intentions. Although research documents that people have various conceptualizations and experiences of transcendence, most empirical studies do not account for the varied and complex nature of the structure of beliefs related to transcendence. In failing to do so, they lose the predictive capacity to identify the salient aspects of transcendent construal. In addition, insufficient recognition of the complex nuances of transcendent construals might lead to missing observable attitudes and/or behaviors in virtues such as gratitude.

What people believe about ultimate reality, the cosmos, or human nature informs how transcendent gratitude is experienced. For example, in a Judeo-Christian context, people often perceive God as psychologically similar to humans, with desires, emotions, intentions, and agency (Nyhof and Johnson 2017), and these perceptions potentially lead people to feel grateful to God in similar ways as they do in regard to persons. Research on transcendent gratitude from this perspective is usually referred to as gratitude to God, and such research focuses on the beliefs and experiences of an agentic, benevolent God (Krause et al. 2015, 2017; Stroope et al. 2013; Wilt and Exline 2022). Cosmic gratitude generally refers to

gratitude to nonhuman sources that are deemed benevolent (Manela 2019; Roberts 2014) or that are viewed as the sources of goodness (Cohoe 2022). However, we must ask, what about belief systems with less tangible benefactors that do not promote a metaphysical personified other? What if no cosmic giver—benevolent, agentic, or otherwise—is recognized? Do they have a similar attribution process?

Given that the prominent conceptual approach to interpersonal and cosmic gratitude involves an appraisal of a gift and personified giver (Cohoe 2022; McCullough et al. 2002; Roberts 2014), we seek a richer understanding of how beliefs factor into gratitude for gifts not attributable to humans, particularly if the giver is nebulous and not easily personified. Additionally, we want to explore if transcendent gratitude is related to prosocial responses in a manner similar to interpersonal gratitude. In order to answer questions such as these, this study explored the nuances of how different transcendent narratives specifically related to experiences of transcendent gratitude.

### 1.4. The Current Study

In the current study, we aimed to broaden current theoretical approaches to different understandings of gratitude. In order to do so, we employed a case-study analysis that was represented by individuals with varied belief systems in order to identify cases that exemplify or perhaps challenge current theories of gratitude. We purposely interviewed individuals who believed in God, those who were spiritual but did not believe in a God, and those who did not identify as spiritual or theistic. A narrative approach provides insight into the substantial features of their lives, enabling us to explore the consistency and coherency of their beliefs throughout their narratives.

## 2. Methods

This case-study analysis is part of a larger mixed-methods study of transcendent gratitude (Mangan et al. forthcoming). The study was designed to elicit person-centered perspectives, as well as variation across and within participants, given their self-identified transcendent beliefs. The following details the relevant aspects of the research design and the data analytic strategy.

The six participants included four females and two males, ranging in age from 25 to 59 ($M = 45.7$) years old. They represent a variety of ethnicities—Black, Asian, Hispanic, and Caucasian—living in both urban and suburban contexts in various parts of the US. The case-study participants were selected from 29 interviews conducted, in which we employed a purposeful sampling methodology to capture diverse belief systems across three categories: theistic, spiritual but non-theistic, and other. The subsample was chosen based on rich and cohesive descriptions of religious, spiritual, and existential beliefs and gratitude, and in order to maximize representation of religious, gender, and cultural diversity.

### 2.1. Instruments

Participants filled out an online demographic and beliefs screener survey (see Procedures below) and participated in a semi-structured interview.

### 2.1.1. Screener

The primary purpose of the first 10–15 min screener survey was to find out more about participants' belief systems to monitor and recruit even numbers of participants for each of our three main belief categories. The screener asked questions about demographics, their work during COVID, and their beliefs. Not only did this information allow us to ensure a distribution of participants across our three belief categories, but it also provided an opportunity for a quality check to eliminate participants who were unable to or did not provide coherent responses to open-ended questions about beliefs. Initially, users were asked, "In a few sentences, what kind of belief system helps guide you?" Then they were asked to choose one word to sum up the belief system they described (i.e., nature, God). They were then given a series of open-ended questions asking about how their belief

systems impacted their lives, their behaviors, their major life goals and informed their decisions, and they were prompted to give examples. Finally, participants were asked to select which category best described their beliefs: theistic (i.e., belief in a god); spiritual but not theistic, or other. The theistic group reflected belief in God. The spiritual but non-theistic group self-identified as believing in something spiritual, such as a force, essence, or power that connects all of life. The other group did not believe in spiritual realities. The other group generally identified as agnostic or atheist.

2.1.2. Interview

In order to gain more nuanced insight into the structure and composition of transcendent gratitude and to understand the meaning of potential benefactors more fully across belief systems, an open-ended qualitative questionnaire was developed. This study utilized a narrative inquiry approach with a lens of appreciation to elicit and interpret participant's narratives and beliefs. Specifically, we used a modified version of *The Life Story Model of Adult Identity* (McAdams 2008). The first section of our interview included key scenes in a life story (e.g., high point; low point; idealized future scene; and religious, spiritual, or mystical experience scene) with added probes about gratitude and joy. The second section focused on personal ideology, faith, and beliefs and drew heavily on the McAdams interview. The third section asked about gratitude, including their beliefs, experiences, and habits involving ordinary, mundane, or common gifts (e.g., good coffee, mail) and also extraordinary experiences of gratitude (e.g., the birth of a child). The fourth section was on joy, and the interview concluded with questions about challenges they had experienced during the early stages of the COVID pandemic.

The narrative approach yielded rich portraits of their lives, including pivotal moments and daily occurrences providing important context for understanding the substance and consistency of their beliefs. The open-ended qualitative approach allowed participants to describe experiences and perspectives relevant to gratitude and their beliefs about transcendence and potential cosmic benefactors.

*2.2. Procedures*

All participants were recruited simultaneously through Mturk and Prolific online data-sourcing platforms. The inclusion criteria for participants taking this study were that they were over 18 years of age; living in the United States; had internet access; and were able to speak, read, and write in English fluently. Participants were recruited through an advertisement to participate in a study on their beliefs and worldviews that would involve an online screener questionnaire, a survey, and a 90-min video interview via Zoom. A total of 976 users saw the recruitment advertisement (81% from Mturk). One hundred and fifty viewers indicated interest and completed the screener (94% from Mturk). Of those 150, 86 participants met our criteria and were invited by email to participate in the study.

Based on the screener survey, 86 participants answered all questions and demonstrated clarity of their belief system and were invited to participate in the rest of the study. Of these qualified individuals, 46 took a second survey on gratitude and were invited for an interview, yielding 29 interviews. Although we had an initial goal to recruit 10 participants in each of the three belief categories, ultimately, the sample size was determined by using the saturation principle, as is customary in qualitative studies. Data were collected and initially analyzed, and at 29 interviews, no new themes were emerging. In addition, throughout the process, participant demographics were reviewed to ensure a near-even distribution of participants' self-identified affiliation: theistic, non-theistic but spiritual, and other.

Data were collected between May and August of 2021. The compensation for interviews changed from $25 initially to $40 and then to $50. We continued to increase the price for two reasons: First, we noticed that mostly homebound people were engaging in the study, so we wanted to make the study more appealing to others. Second, given the

involved nature of the study, we had a difficult time recruiting new participants, so we increased the incentive to bolster participation.

### 2.3. Data Analysis

Although qualitative studies often seek nomothetic knowledge by identifying themes common to participants and potentially applicable to the populations of interest, our goal was to seek idiographic data and focus on the particularities of individuals around their understandings of meaning and experiences within the context of their lives (Allport 1965; McAdams 2011; Schachter and Ben Hur 2019). We sought to understand the thoughts, feelings, and behaviors of gratitude for specific individuals in a holistic and person-centered manner to preserve the gestalt of participants' experiences and to ground the understanding of gratitude within the contexts of their particular beliefs and life narratives.

For the purposes of providing examples or challenges to existing theories about the structure of cosmic gratitude, we required that the interviews provided rich, coherent, and consistent descriptions of (1) their cosmic beliefs and how they made sense of the world; (2) to whom or what they were grateful; (3) their behaviors associated with gratitude; and (4) their attitudes, feelings, or perspectives that accompanied or were in response to gratitude. A team of four researchers reviewed the 29 interviews in order to select two interviews from each category that represented distinct belief systems. The interviews were reviewed by at least two researchers and then discussed with a broader team of five to reach a consensus on the interviews that represented diverse and coherent perspectives. In the iterative process of reading and distilling each interview or case study, we sought to personalize the analysis by abridging the interview narratives in a manner that retained the whole-person autobiographical tone while noting both personal and ecological particularities that distinguished their beliefs and conceptualizations and descriptions of gratitude (McAdams 2011; Schachter and Ben Hur 2019).

## 3. Results

For each participant, we provide a brief narrative description and offer quotations from him or her to highlight beliefs and experiences surrounding gratitude.

### 3.1. Theistic

Within our theistic category, we chose two participants who clearly delineated a belief system about a metaphysical being, specifically God.

#### 3.1.1. Jennifer

Jennifer, a 46-year-old Black female, reported living with chronic pain and other medical issues. Her husband, with whom she had a good relationship, died of pneumonia in 2014. She self-identified as theistic but did not associate with any particular religion. However, she noted that she believes in the Holy Bible and prays to God. Jennifer's beliefs were influenced by her parents and her late husband.

For Jennifer, she said that her "guiding system of what to do and what not to do in life" was rooted in Christianity. She said she tries to "treat people the way they want to be treated, [and] to not still do things that harm others". While she reported at the time of the interview that she did not attend a specific church, she proactively engaged in practices such as reading the Bible and listening to sermons on a weekly basis, as well as praying daily. She constantly experienced God's comforting presence, saying, "He's always there. He's always with me . . . giv[ing] me a sense of protection. Yeah. A sense of comfort. Yeah, yeah, it's very comforting to know God's there. Always". She described God as having attributes such as being a protector, compassionate, loving, caring, helpful, and supportive.

She spontaneously linked her beliefs about God to gratitude. In response to a general question regarding her religious and/or spiritual approach to life, Jennifer immediately replied: "I believe in thanking God, being grateful, showing gratitude, being grateful for

what I have, [I] try not to complain about what I don't [have], and helping when I can, helping people, giving back".

God's role in her ideas about gratitude is evident when she described paying for her husband's memorial service. She recalls the following:

> When I had to pay for his service, memorial service, I didn't have enough money. And I needed help, and a co-worker of his started a collection and took up the money. I had been praying [for] an answer, and to me I believe God answered that prayer. Because we took up a collection and I was able to afford to pay the funeral home for the service. To me, that was God. God, because I didn't know, and I put it in God's hands, and he answered . . . I couldn't have been more grateful, I was truly grateful for that.

She then summarized, "I was grateful for God for answering the prayer, for God sending the people, for the people that donated, I'm just grateful. All of that". Additionally, when asked about what she was grateful for, Jennifer easily listed the following: "My husband, our relationship. I've been grateful for people that helped me along the way. I've been grateful that my parents are still here in the land of the living, relatively healthy. I've been grateful that medical procedures have gone well. I've been grateful for having a roof over my head, food to eat, having enough money to pay bills".

She also was clear about behaviors connected to gratitude. She expressed her thanks to others, and she said prayers of thanksgiving to God. At one point in her life, she kept a gratitude journal. "I did a daily journal, and I got the idea from Oprah. Did a daily journal of expressing, what is it? Five things [of] gratitude a day. Find five things you're grateful for each day, and write it down in a journal".

### 3.1.2. Randal

Randal, a 44-year-old Caucasian male, was living in a suburban environment in the Pacific Northwest at the time of the interview. He worked in data analysis and reported being married to his wife for five years. Although he grew up in the Pentecostal Holiness tradition in a remote, fairly poor agricultural community in Central California, he is now Eastern Orthodox. Early in his life, he began to value nature and the interdependence of life, but these views became more established through the Eastern Orthodox community.

Randal attributed a number of personal attributes to God: "God is good. God is loving. God is patient. God is gentle. Does not force himself. God is as pained as we are". He engaged in spiritual practices with his community, such as weekly liturgy, taking communion, daily prayer, and confession, which connect him to God. He valued the energies between community members and how that exchange reflects back to God. He explained, "So basically, God loves humanity and wants to raise us up into his Kingdom. And His energies permeate the world and permeate us and all of our relationships with each other and with him . . . it's part of my belief structure that God is in everything and everything is interconnected". He highlighted the importance of what this belief meant for him practically: "So knowing God is experiencing those energies, and coming into knowledge of our interdependencies with each other, our love that binds us all together, and how that then reflects back up to God".

Randal reported feeling gratitude for the beauty around him, as well as the beauty he saw in people when they treat one another with love. When talking about a transcendent giver, Randal ultimately attributed all gifts to God: "It's the story in our worldview that God is reconciling the world to himself . . . all of our good gifts come from him". He felt most grateful for beauty, love, and interconnectedness: "When I'm really impressed and feeling most grateful . . . It's when I see that people are just in sync with the way that things should be. That they're living in love, that they're living in beauty, that they're living in grace . . . . That's the deepest gratitude. It's when you see a life being lived well". Additionally, Randal highlighted the importance of being grateful for everyday life: "I'm not highly emotional, but when I see it, it's like I recognize it and I appreciate it. It's like smelling good food". In reference to interpersonal gratitude, he reported thanking and

acknowledging people when appropriate or keeping it to himself and smiling. He also said that much of his gratitude is directed to God.

### 3.2. Non-Theistic but Spiritual

Within the second category, belief systems varied broadly. Narratives not only included a variety of beliefs (e.g., magic, God), but beliefs ranged in degrees of consistency and comprehensiveness. For our purposes, we chose two participants with distinct non-theistic belief systems that were coherent and linked to their experiences of gratitude.

### 3.2.1. Esther

Esther, a 58-year-old Caucasian woman, was living in a large urban context in the Midwest at the time of the interview and was an artist and sculptor. Esther was the youngest of seven children and noted her early love of nature and the wilderness that brought her peace. She referred to herself as a "very spiritual" person. She reported growing up in Catholicism but questioning the existence of God amidst tragedy.

Her beliefs became shaped by readings on Buddhism, Taoism, philosophy, and science and her connection to nature. She reported that these beliefs center on an awareness of an "essence" she referred to as "life" which can be found in an awareness and connection to life, nature, and others. She explained, "I'm very nature-based; I do believe in, that there's essence of what I refer to as life, which isn't the heart beat or anything, it's more an awareness, I believe that it's in everything natural, not the unnatural things that we create, but that it exists in everything natural, and we should respect it".

When asked about what it meant for her to be spiritual, she explained that her body is only a vehicle to interact and connect with others. "To me, it means realizing that this is not me, this (body), it's just a convenient little bottle that I'm stuck into so that I can interact and people can interact with me. I believe our thoughts have just as much of an impact on all of life as we know it as our actions do, we feel negatively, we are contributing to that negative". She offered very vivid descriptions of how her meditation practice connected her to this energy: "I believe that by asking, like when I meditate, by opening my mind and being receptive to what is good and what is right, that I'm actually drawing that towards me, almost like lighting a candle in the dark in order to draw our modes in, you're becoming magnetic towards it".

However, for Esther, this source is not personified: "I just see it more as energy output, and by sharing, we multiply it, and that's what keeps it going". Her beliefs about the interconnectedness of all living things and her commitment to being aware were evident in her reflection on decision making: " . . . whatever I do to try, and I think consciously, 'Why am I doing this? . . . Is it the right thing to do? Is it for good, or is it selfish? . . . And try to focus everything I do on having the best impact, and especially when I'm dealing with other people, animals, that I can just think of what is best for them at that point in time".

In describing gratitude, Esther explained the following:

> I think gratitude is truly and completely recognizing the gifts that we are presented with, not from people and all, but from our experiences in life, and showing respect by actually recognizing them and feeling thankfulness and meekly, that what we're given is nothing that we could have achieved ourselves, that the feeling for love, just recognizing when you have an eureka moment, you suddenly understand a situation, gratitude is just a deep feeling of recognition, respect, and thankfulness for an experience that was given to you.

She further offered graphic descriptions of how gratitude served to energetically connect and benefit all things:

> I imagine, say, I'm interacting with you and I really feel much gratitude for you, I imagine more like if you had a spectrometer or something and you could see the auras that we put forth or the energies that we feel, it would come from me, it would just wash through you . . . then from there it multiplies and goes out into the essence, into the universe, the energy that goes through all of everything.

She provided several examples of gifts of gratitude. She was thankful for sharing her life with the people whom she loves; feeling wonder about nature, the sun on the trees, the clouds, and the rain; and the opportunities that each day brings. Esther reminded herself daily of the things she is grateful for. She reported these experiences as being linked with emotions of awe, being overwhelmed, and feeling a sense of humility. When she reflected on how she practically responded to gratitude, Esther reported that, when she was alone, she would smile, laugh, and say "Thank you" quietly to herself. When with others, Esther says she expressed her gratitude out loud, perhaps by taking a moment to stop, breathe, and express her thanks.

### 3.2.2. Maria

A 42-year-old Hispanic female, Maria, was a single mother, raising teenage children. She grew up in a low-socioeconomic neighborhood in a small city in the Dominican Republic. When she was a teenager, she immigrated to the United States with her father. Maria reported placing a high importance on dreaming and working hard for her aspirations. She grew up in a culture where "men dominate[d]", but she strayed against the norm and decided to pursue a career in Accounting and Software Engineering. She stated that she lives in the United States in order to provide her children with more robust educational opportunities.

Maria grew up Catholic in a country where 90% of people are Catholic; however, she reported no longer attending church because she felt she could pray and read the Bible on her own. She talked about how her beliefs evolved:

> I believe a lot in the cause and effect. You do something, there's a consequence to everything that we do, good or bad . . . And then personal responsibility. I get annoyed when people think that everything that happens is just luck or someone else's fault and doesn't want to take their own . . . responsibility for their own life . . . And that's my main belief. I think I'm responsible for everything that happens bad in my life, and for a certain percentage of the good things that happen, [chuckle] too, because there are good things, there are other people that intervene in the good things.

She also described a force that has supported her: "And probably because the way my life has been, I know there has been (a) certain power, force that helped me, people that I met that helped me, that supported me". In her own life, Maria experienced this power helping her to navigate tough life situations: "You have five good years, and then you have a couple of bad ones. But I feel like it has been there in the bad years as well. And somehow, it has helped me navigate, go to the good years in one piece [chuckle]".

She reported feeling empowered through this force to stay strong and to keep her mind strong, driving her to continue to "work to change it": "And it has been there, that force, it has been there to help me with my mind . . . " She described this power as dependable and fair: "But in my life, when I feel about these attributes of the higher power, I can tell you, to me, is something that is, how can I say it, fair, very fair for my life. Like I do my work, I feel I get reciprocated, so it's a fair power. It's dependable . . . "

Transitioning to Maria's perspective on and experiences of gratitude, Maria reported feeling grateful for her children, for being able to take care of her family's emotional and financial needs, for coffee in the morning, and for being able to work from home during the pandemic. When she was asked about the high point of her life, she described her deep joy and gratitude for being able to help a relative pay for college and how this gratitude translated to a deep desire to give back to others:

> I cannot quantify the value of that amount of money that I gave her, because right now, it translated exponentially. She's a professional at teaching . . . Those four or five years changed everything . . . just to be part of that, that was . . . I felt like it was one of my kids that was graduating from college, to be honest. So that one that was profound to me, and I felt at that moment, that I was . . . That was a gesture of gratitude to life. That person didn't do anything to help me

anywhere, but to me, represents a lot of other people that helped me throughout the way. A lot of things that get combined within my life to help me. And I feel that obligation, and it's not like an obligation, but I feel [the]thing that I need to go . . . To give back because I just received so much.

This feeling propelled her forward, motivating her to continue to help others. "I feel so good, but also I felt also compelled and responsible, maybe to do more in the future with other people". This sentiment of experiencing gratitude for good gifts in her life which prompted her to pursue prosocial behaviors is summarized well in this final quote:

"I think it's, to me, is just simple, to be aware of the good things that has happened in your life and to be able to get back, to give back, not necessarily to the people that probably helped you in the moment, because sometimes maybe those people are not around or you just can't give back to them, but give back in general to society, to other people that you know".

### 3.3. Other

The third category includes participants who identified as neither religious nor spiritual. We selected two case studies where the individuals clearly articulated their belief systems and how those systems played out in their lives.

### 3.3.1. Tina

Tina is a 59-year-old Caucasian female who, at the time of the interview, was living in a suburban environment in the Southeast. To bolster her strength, in her twenties, she joined a Medieval reenactment group and learned to fight in a full suit of armor. Tina's worldview and morality were shaped by the code of chivalry promoted within this group. She moved beyond that association, but it was formative for her. More recently, she became a competitive cyclist.

When interviewed, she said that she enjoyed trail running, cycling, and mountain biking. Tina stated she was not even "remotely religious". When asked about her spiritual beliefs or beliefs in God or a Higher Power, she flatly responded, "No". Despite this response, she described transcendent experiences that connected her with nature and with others:

"I certainly get overcome by a sense of wonder when I do star-gazing . . . It's like a little 30-s thing of just complete happiness of being able to observe these things that happen in nature . . . there'll be like this big open field with a bunch of those little yellow flowers and I'll just stop and be like, 'Whoa'. Or like right at sunset when everything sort of turns like a pink color sometimes, it never ceases to amaze me".

She reported feeling a deep bond with other mountain bikers that contributes to a "pay it forward" mentality.

Somebody was so cool to me on the trail and they helped me when I was lost. I mean, I've literally had people I don't even know see me go out on the trail after sunset and wait to make sure I got back . . . So when that happens, I wanna pay it forward . . . I take this little mental Rolodex, like, 'I gotta do something nice for somebody else. I gotta keep this moving'. And that's kind of a compulsion[as] . . . I absorb all this feeling of wellbeing, then I wanna make sure I share that, I wanna do a nice thing for somebody else, it's almost like somebody did me this huge favor, and this is too much for one person. I gotta make sure and help somebody else.

Tina reported a belief system that was deeply informed by an engrained moral code of chivalry that she picked up while participating in medieval reenactments:

"That whole kind of personal code of chivalry that I took away from that, I apply it all the time. And it sets the bar for human behavior unreasonably high, but I

keep trying. 'Cause it's basically expecting you to be the perfect individual, and nobody is, but there's nothing wrong with continuing to strive for that".

Tina said gratitude was "appreciating what you have in life to some extent, I guess, material lives, but just experiences. Yeah, it's just appreciation and recognizing what good things are in your life and being happy about that". Tina reported being grateful to a wide variety of friends and family, and she actively pursued behaviors to express these sentiments: "Well, if it's my husband, I'll just text him and tell him how awesome he is . . . I'll go, 'God, he's got so patient, he's so great'. I tell my older sister a lot, 'cause I'll call her and I'll just be freaking out about something, or she'll even call me, which makes me feel like really useful . . . "

Although Tina reported that she does not believe in a cosmic realm, she holds tightly to a set of beliefs that guide her moral behavior. In addition, she was able to experience transcendent connection with both others and nature, both of which seemingly affected her practice of gratitude. She was able to quickly identify those she is grateful to and was able to note both the prosocial behaviors and emotions experienced (for example, awe and wonder).

3.3.2. Ben

Ben is a 25-year-old Asian male who, at the time of the interview, was living in a suburb in New England. He reports being a part of a community of gamers. Ben's mother was Buddhist, and he remembered partaking in some Buddhist practices, such as praying and burning incense, while growing up. Ben reported meditating daily before bed to promote a sense of calmness. He identified as agnostic.

When describing his beliefs, Ben said, "I believe in what I know and what I observe around me and what others have observed and know. That's what I base my beliefs on . . . .As for anything else, I mean, I have an open mind about it. I leave it behind everything that has been proven, and it can be proven, and that's basically my belief system". As for Ben's sense of morality, he considered whether something "has a bad effect on other people".

Ben reported that it is important for him to be informed and make informed decisions, and he relies on what is observable or provable. When Ben reflected on gratitude, he explained that people should be "grateful for something that you have received at the expense of someone else". He also noted that you "can be just grateful for your situation, 'I'm alive, wild nature is beautiful and all that. I'm glad to be on this Earth'. We didn't really provide the Earth to ourselves . . . I am grateful that Earth exists. Thank you, Earth for existing".

When he detailed his feelings about gratitude, he compared himself to people less fortunate and said that he was grateful for the following:

> The shelter I'm in right now, computer that I'm using to conduct this interview, I am living in luxury. I know that a lot of people are not living in this sort of luxury every day. I'm grateful for just being in the situation, I was born here, I never had to go out and hunt for food . . . I could just live here and I can go out and protest for something stupid and not be punished for it. It's a great place to live, as far as I can tell.

When Ben was asked to think about extraordinary experiences of gratitude, he struggled to find a response and then said, "Oh, heck no . . . !" Ben did not report a gratitude practice, and when asked about potential behaviors resulting from moments of gratitude, his comments came off as hypothetical and not very thought through. He explained, "If you're grateful for nature, just it being there, I *guess* you would pay it back by being more conscious of . . . let's say the products you buy, it has to be environmentally friendly or something. You could pay it back by just swapping out a few product brands, make sure that they're ecologically responsible". When speaking about his parents, he said, "I'm grateful for my parents for taking care of me, Mother's Day and Father's Day exists for a reason. You can also keep track of their birthdays if you're especially attentive to their

needs. Get them gifts and stuff, say the mushy, mushy 'I love you, Dad and Mom' and all that. We're not mushy, mushy though. Definitely not. A lot more business".

## 4. Discussion

Given the interest in exploring gratitude to transcendent or other non-human sources, and the shifting trends in religious and spiritual affiliation (Smith 2021), we aimed to explore how different belief systems impact gratitude experiences and their related behaviors. In order to capture the nuances of beliefs, we conducted a case-study analysis of six individuals with different worldviews. A life-narrative approach combined with semi-structured questions about gratitude allowed us to investigate the relationship between beliefs and gratitude across three different belief categories: those who were theistic or reported believing in God, those who described themselves as non-theistic but spiritual, and those who identified as other. We wanted to explore how individuals conceptualized and experienced gratitude for things that were not attributable to humans. We hoped to broaden the prominent conceptual approach to gratitude that emphasizes recognition of a gift and response to a benevolent, agentic giver as the fundamental appraisal processes underpinning gratitude (Cohoe 2022; Manela 2019; McCullough et al. 2002; Roberts 2014; Tsang et al. 2021). In addition, because the research demonstrates that interpersonal gratitude enhances relationships and generates prosocial action, we investigated the narratives to see if various conceptualizations prompted the same prosocial benefits.

Guided by the conventional understanding of gratitude, we examined whom or what each participant considered to be the source of gifts, to whom or what they thanked for these gifts, and their attitudinal and behavioral responses to those gifts. In all instances, the six participants were able to articulate to whom or what they were grateful. However, not all participants were able to name or describe a transcendent or non-human source; rather, they defaulted to expressing gratitude to persons or to the gift itself. Although the participants varied greatly in their beliefs and conceptualizations about gratitude, in each of the six cases, their narratives provided an arc of life experiences, transcendent beliefs, and views and experiences of transcendent gratitude.

For Jennifer, God was personal, a loving God who provided personal gifts directly to her or through others. She very intentionally prayed to and thanked God personally. In contrast to Jennifer, Randal's narrative emphasized the interconnectedness of all. He viewed God as loving and permeating all of reality. He was grateful to God for gifts that benefit all (e.g., beauty) and when people lived in harmony and grace. Both theists emphasized the importance of thanking God directly for gifts. Similar to Randal, Esther valued the interconnectedness of reality and a life essence. Although she did not name a cosmic giver, she described gratitude as an opportunity to intentionally send energy to the giver and beyond to the essence of life. Maria, a former Catholic and daughter of immigrants, valued personal responsibility. Although she recognized a higher power, she felt accountable alone for what she had earned. She recognized that others, such as her parents, helped her, so she desired to pay these gifts forward and assist others on their journeys. Tina also valued paying goodness forward. An explicit atheist, Tina did not recognize a transcendent source of any gifts. That said, she was animated by transcendent emotions of awe, wonder, and profound happiness when she found herself in nature or was helped by others. She had no cosmic giver to thank but felt a compulsion to pay gifts forward. Conversely, although agnostic, Ben's beliefs and statements of gratitude were directed toward what he observed. He viewed gratitude as something necessary when there was a perceived cost to another, and he viewed it as obligatory to express thankfulness by taking care of the gift.

These idiographic findings exemplify and challenge the existing theory and research on gratitude that identify as necessary the acknowledgement of a gift and an agentic and intentional supernatural giver (Cohoe 2022; Manela 2019; Roberts 2014). The two theistic participants, Jennifer and Randal, provided fitting examples to the existing gratitude-to-God literature (Krause et al. 2015; Roberts 2014; Rosmarin et al. 2011). They showed how

people construe an agentic and beneficent transcendent giver. They both expressed thanks directly to God. Their narratives correspond with existing theories such as Algoe's (2012) find–remind–bind theory. For both theists, the recognition of gifts prompted them to find God, the giver of all good things. The gifts, even when given through people, reminded them of their ultimate source, God; bound them further with God; and increased their gratitude and closeness to God.

The other cases challenge the prominent exchange approach to gratitude. Debates within the philosophical literature argue that in order to experience gratitude even within non-theistic worldviews, individuals need to have a benevolent giver to thank (Manela 2019; Roberts 2014) or perceive a source of goodness (Cohoe 2022). In addition, the psychological literature emphasizes that the cognitive appraisal of a good giver contributes to the relational and generous outcomes associated with gratitude (Algoe 2012; McCullough et al. 2002, 2008). That said, the remaining four cases, whether the individual identified as spiritual or not, did not report agentic givers deserving of gratitude or thanks. Furthermore, three of the cases articulated beliefs about paying gratitude forward to others, whereas the fourth, Ben, articulated the duty of paying back and sustaining the gift. For the three, in their own unique ways, they articulated how gratitude involved the recognition of a gift from a beyond-the-self source, and sometimes from beyond another human agent's credit, providing the opportunity to invoke reflection and awareness of something more than one's immediate life. For Esther, gratitude offered an opportunity for appreciation and respect that prompted meekness and humility, stirring her desire to contribute to the greater essence of life. Tina described being overwhelmed with the positive emotions of delight, awe, and wonder and being compelled to pass on these feelings of well-being to others. Maria, who most valued agency, was grateful for those who furthered her efforts and was deeply delighted to help others. Although her "paying it forward" might have originated from her quid pro quo mentality, she recounted that the high point in her life was the joy of helping someone graduate, and this experience inspired her to continue to help others.

None of these participants described a worldview with an agentic or benevolent, transcendent source of gifts. Regardless, emotionally stirred by their awareness of gifts, the three offered goodness forward. Although the literature suggests that gratitude might be easier to access if individuals believe in a personified transcendent source (Tsang et al. 2021), these findings illustrate that gratitude is possible to experience not only for non-theists, but also for those who have no explanation of a transcendent source. In particular, the interviews of Esther, Maria, and Tina suggest that an ultimate source of gifts is not necessary for gratitude to occur. Rather, their responses to gratitude were directed toward tangible acts to help others or energetically to the universe. Although technically or philosophically, some might describe their responses to gifts as *appreciation* because they had no giver to thank directly, appreciation does not account for their conscious responses of paying it forward. Perhaps, these cases challenge the proper understanding of gratitude that is dependent on thanking the giver. Furthermore, these narratives also challenge existing assumptions that gratitude produces prosociality because it alerts individuals to the prosocial efforts of others, prompting them to behave with the same prosociality (McCullough et al. 2001, 2008). In the case of these three participants, they had no prosocial (or otherwise) giver to be alerted to, yet they were still prompted to act generously toward others. Their examples may offer a modification to Algoe's (2012) find, remind, bind theory of gratitude for those whose belief systems have no transcendent giver. For the three women in this study, the find–remind–bind theory might be explained by finding a gift, not necessarily a giver; being reminded to appreciate that it was not of their own doing; and binding with others by passing on a gift.

Although the motivation or impulse to respond as part of gratitude—whether giving back or paying forward—might be explained through the broaden-and-build theory of positive emotion that suggests that positive emotions open our perspectives and motivate us toward action (Fredrickson 2004), the direction toward which the action might benefit

is not predetermined. The good feelings that one experiences, whether like Tina when she experienced help while mountain biking, Esther when she experienced connection in nature, or Maria when she delighted in helping another, might spur one into action, but they do not guide behaviors directionally. Might the cognitive appraisals pertaining to gifts from sources beyond one's own doing, even without understanding or acknowledging a source of the gift, prompt enough of a beyond-the-self awareness and possibly attitude to stir generous and beyond-the-self actions? Furthermore, comments in Esther's narrative that named meekness and humility call into question whether receiving a gift from a beyond-human cause might give pause and reason to consider one's significance and, thus, responsibility.

An additional potential explanation of the prosocial responses to gratitude by individuals whose worldviews indicate no clear transcendent giver may be found in socio-affective neuroscience. Research by Immordino-Yang (2016) suggests that when people reflect on abstract beliefs and simultaneously experience transcendent emotions (e.g., gratitude and awe), they internalize the beliefs as meaningful (see also Riveros and Immordino-Yang 2021). Although we have no brain images of these participants, their narratives offer descriptions of transcendent emotions—being overwhelmed with awe, with wonder or experiencing mystery, and connecting these feelings to experiences that they internalized as beliefs that direct/orient their lives. For example, Tina recounted being overwhelmed by amazement when people assisted her while fighting or racing and that she was then compelled to pass on the feelings of well-being to others. Even Maria, who viewed the world in terms of cause and effect, was so moved by the joy of helping another make progress in her journey that she committed to continue to help others.

Ben provides an important contrast. He described himself as unemotional. He reported that his family was "not mushy", but "all business". His life was confined to gaming and possibly limited in regard to human interaction. Not surprisingly, his statements about beliefs and gratitude were similarly flat. His perspective was limited to the observable and what had been provided, and his narrative suggests that his life was very limited. Given his young age, the confines of quarantine, and that he reported spending the entirety of his time alone on his computer, either gaming or working through MTurk, his limited reflections on his beliefs and views of gratitude are not surprising. He reported being grateful for "what is", and recognizing a cost or a debt. His lack of emotion corresponds to his lack of enthusiasm and appreciation for anything beyond what is observable and obvious.

In addition, Ben's narrative contrasts to the others regarding the link between gratitude and connection to others. Ben's narrative had few mentions of relationships, except for saying that his were "all business" and describing a hypothetical sense of obligation to his parents to give them Mother's and Father's Day cards. He did not speak of friends, colleagues, groups, or organizations that he engaged in. He had no sense of transcendence–metaphysical or otherwise. Similarly, his descriptions of gifts were limited to things provided to him by others. He did not describe any prosocial or beyond-the-self inclinations. Gifts were viewed as a cost to others; there was no recognition of loving or kind intentions. The expression of thanks to the gift was a duty. There was no evidence of his being inclined toward sharing or paying his blessings forward.

Taken together, our findings suggest that beliefs do not need to involve a metaphysical other to provide meaning and substance to a transcendent narrative identity and to inform the meaning of virtues such as gratitude (King et al. 2020; Schnitker et al. 2019). These case studies provide six unique examples of how beliefs and worldviews inform gratitude and its perceived obligations and responses. These narratives exemplify how transcendent beliefs, whether theistic or not, are relevant. Five of the cases reported a clear sense that there was something more than themselves, and gratitude resulted from beyond-the-self realizations, prompting individuals to give thanks to God, offer good energy to the universe, or be helpful and generous to others. The one participant, Ben, who indicated no sense of transcendence, provided a stark contrast insofar that his lack of beyond-the-self (in his case beyond his home and computer) beliefs or experiences were consistent with his inability to

articulate gratitude beyond his immediate current situation and his lack of articulating any actual beyond-the-self prosocial actions.

The varying experiences of transcendence reflected in these case studies are further understood in light of the distinctions between ontological, phenomenological, and subjective transcendence (Callaway et al. 2020). Jennifer and Randal's experiences of gratitude to God exemplify ontological transcendence relating to a metaphysical other. Whereas Esther's experiences reflect phenomenological transcendence insofar as her explanation of transcendence was less predicated on beliefs but more on the experience of expanding beyond the self and feeling connected to the life essence. Tina and Maria exemplify different examples of subjective transcendence. The emotional resonance and stirring of emotions within Tina's narrative suggest an experience of transcendence which prompted generous actions. Maria emphasized personal agency and effort, and she also reflected that she found meaning through her efforts and connections to help others. Ben's narrative reflects Callaway et al.'s notion of an isolated self that may or may not perceive or experience transcendence.

Although the existing research suggests that transcendent gratitude involves the receipt of a gift, recognition of a giver, and a response, our findings suggest that, within worldviews without agentic sources of gifts, transcendent gratitude may involve appraisals of receipt of a gift, recognition and respect that there is more beyond the self, and prosocial responses.

### 4.1. Limitations and Future Research

Although these case studies provide textured nuances of beliefs in the context of details of actual lives, allowing for a thorough investigation of less conventional belief systems and gratitude, there are limitations to this approach. Important for consideration is the fact these narratives represent only the perspectives of the participants. We have no other sources of data to vouch for their credibility and validate that the attitudes, experiences, and behaviors have actually transpired. Although these six participants' interviews represented self-effacing comments and seemed genuine and authentic, research demonstrates that, in research, some people present themselves as socially desirable to a researcher and over-report qualities that may be deemed as good (Krumpal 2013). Additionally, our data were collected at a unique point in time, so we do not have a way to test if their responses would be reliable over time. Additional research would benefit from a follow-up study with the participant and include an additional source of data, such as a friend or family member.

It is important to note is that the data were collected from May to August 2020, which was at the height of the COVID-19 pandemic, with most of the United States living in quarantine, and when no vaccines existed. At this time, people around the world were still adjusting to the threat of the disease, possibly dealing with losses of loved ones and jobs, living in social isolation, and the unknown. Lives were marked by disruption, and mental health issues began to surface with enduring agitation and adversity. For some people, times of adversity and uncertainty become opportunities to reconsider and recommit to religious and spiritual beliefs. Additionally, such circumstances can intensify a sense of gratitude for life (Emmons and McCullough 2003). Consequently, we do not know the implications of the COVID context for this study, but we recognize that these interviews were conducted during an extraordinary moment in history.

Although these participants self-selected into the study, the case studies were chosen deliberately from the original 29 for the coherence and consistency of their beliefs. We recognize that some of the other 29 in the broader data set did not offer belief systems that were as rich or comprehensive, thus making it difficult to connect beliefs to gratitude. So, although our sample and findings expand existing conceptualizations of cosmic gratitude, they may not be relevant to all persons, even for those who share similar beliefs. For example, one participant who self-identified as not theistic but spiritual was not included in our analysis because he said he was agnostic, but still used God as a referent, and expressed gratitude to God. Several participants described that they rejected the conventional beliefs

of their youth or families but indicated that they did not fully differentiate a belief system. Further research is needed to explore how transcendent gratitude transpires for people who have ideas that are less clear about how they construe reality, how they make sense of their lives, and what holds ultimate value.

Although these findings challenge existing conceptual frameworks of transcendent gratitude that argue for a delineated target of gratitude (Cohoe 2022; Manela 2019; Roberts 2014), they are only initial findings and focus on belief systems. Given that we were curious about people's beliefs about potential targets of transcendent gratitude, we focused on their beliefs and not individual psychological differences. The narratives suggest that the participants varied in terms of their emotional range and regulation, the extent to which they related to others, and possibly their levels of dispositional gratitude that may all influence their capacities for transcendent gratitude. For example, the two men in our study, Randal and Ben, both described themselves as less emotional, and they either lacked or had the least developed descriptions of inclinations toward paying gratitude forward. In contrast, the four women had quite effusive descriptions of transcendent emotions, such as awe and wonder. In addition, they demonstrated strong self-awareness and spoke more vividly about their connections to God, nature, and/or other people. These differences raise questions such as, does one's attachment style or emotional-regulation tendencies influence one's experience of transcendent gratitude? Does attachment style or emotion regulation factor in differently if one has a more personified or a vaguer notion of a transcendent giver? Further research is warranted to explore and evaluate if different psychological tendencies may be more apt for different forms of cosmic gratitude.

*4.2. Conclusions*

These case studies illustrate how people with various transcendent belief systems are able to have psychologically rich experiences of gratitude for gifts that are not attributable to humans. Even though they may not recognize an ultimate source or transcendent giver to thank, they still are prompted to respond to gratitude by sharing goodness with others. The appreciation for gifts may prompt beyond-the-self thoughts and emotions that prompt future beyond-the-self actions. Participants' descriptions suggest that transcendence does not hinge on metaphysical beliefs or deliberate appraisals of a benevolent benefactor; they can be experienced through a sense of interconnectedness or union with life, nature, or humanity. Alternatively, others transcend themselves through profound emotional resonance or a sense of agency experienced through themselves. These findings suggest that current approaches to gratitude that are dependent on appraisals of a gift and giver may overlook viable opportunities to experience gratitude and related prosocial outcomes.

**Author Contributions:** Conceptualization, P.E.K. and R.A.B.; methodology, S.T., S.A.M. and P.E.K.; software, S.T. and S.A.N.; validation, R.A.B., S.A.N. and S.R.C.; formal analysis, S.T., R.A.B., S.A.N. and S.R.C.; data curation, S.T. and S.A.M. writing—original draft preparation, P.E.K., R.A.B., S.A.N. and S.R.C.; writing—review and editing, P.E.K., R.A.B. and S.A.N.; supervision, P.E.K. project administration, P.E.K.; funding acquisition, P.E.K., S.A.M. and S.T. All authors have read and agreed to the published version of the manuscript.

**Funding:** John Templeton Foundation 61513.

**Institutional Review Board Statement:** Human Subjects Review Board, Fuller Theological Seminary.

**Informed Consent Statement:** Informed consent was obtained from all subjects involved in the study.

**Data Availability Statement:** Please contact first author.

**Acknowledgments:** The authors would like to express their gratitude to Jill Westbrook for her invaluable feedback on this article and to Lauren Van Vraken for her assistance in coding.

**Conflicts of Interest:** The authors declare no conflict of interest.

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
