# Peer review of "Shades of Gratitude: Exploring Varieties of Transcendent Beliefs and Experience"

_religions, doi:10.3390/rel13111091_

Round 1
Reviewer 1 Report
My appraisal of this article is more than positive. The topic is interesting, the purpose significant and methods employed are fresh and efficient. The content reflects the efforts taken and a time-consuming work when article was prepared. It is also well-written and well-composed.
One remark: the sentence in rows 169-171 is probably lacking the verb.
All in all, I recommend the article for publishing.
Author Response
Thank you for your kind comments. We addressed the suggestion and added a verb to the sentence in rows 169-171..
Reviewer 2 Report
Considering the conditions under which the data were collected, it is a well-documented and elaborated qualitative work.
Author Response
Thank you!
Reviewer 3 Report
This is a very interesting paper and I would recommend to publish it
I have some recommendations for minor revisions;
1.) I could not find "Padgett, 1998" among the references.
2.) The distinction between ontological, phenomenological and subjective transcendence is very interesting, but in my opinion the distinction between phenomenological transcendence and subjective transcendence needs some more clarification.
3.) I have some problems with the distinction between three groups of participants, the theistic ones, the non-theistic, but spiritual ones and the others. The distinction between theistic and non-theistic belief systems is very clear and relevant, but the group of non-theistic, but spiritual participants is not clear to me. The label "spiritual" does not appear in the narratives of the "spiritual" participants and I do not understand the relevance of a distinction between people with a non-theistic, but spiritual belief system and the others. Perhaps the authors can clarify this distinction a little bit better than they have done in this article.
Author Response
1.)We eliminated the "Padgett, 1998" citation.
2.) The distinction between ontological, phenomenological and subjective transcendence is very interesting, but in my opinion the distinction between phenomenological transcendence and subjective transcendence needs some more clarification. We attempted to clarify the definitions and distinctions of these terms in the paragraph starting on line 66.
3.) I have some problems with the distinction between three groups of participants, the theistic ones, the non-theistic, but spiritual ones and the others. The distinction between theistic and non-theistic belief systems is very clear and relevant, but the group of non-theistic, but spiritual participants is not clear to me. The label "spiritual" does not appear in the narratives of the "spiritual" participants and I do not understand the relevance of a distinction between people with a non-theistic, but spiritual belief system and the others. Perhaps the authors can clarify this distinction a little bit better than they have done in this article. We elaborated on the distinction of the Spiritual not Theistic group from the Theistic and Other groups in the "Current Study Section" (lines 188-191) and in the "Methods/Participants" section (lines 230-233).
Reviewer 4 Report
Point 1: On line 94, you note that gratitude is a state and trait; please explain what the difference is between gratitude as a state and gratitude as a trait.
Point 2: On line 129 you lump karma within conceptualizations of the transcendent. Is this accurate? Karma literally refers to causality, specifically the moral consequences of our actions, which means that karma is generated here and now.
Author Response
Thank you for pointing these two issues out.
Point 1: On line 94, you note that gratitude is a state and trait; please explain what the difference is between gratitude as a state and gratitude as a trait. We added a brief explanation for state and trait on lines 116-117.
Point 2: On line 129 you lump karma within conceptualizations of the transcendent. Is this accurate? Karma literally refers to causality, specifically the moral consequences of our actions, which means that karma is generated here and now. Thank you for pointing out this error. We took karma out of this discussion of transcendence in that section.